# BLCSRec: Bridging Language and Collaborative Semantics for Next POI Recommendation

## Abstract

Next Point-of-Interest (POI) recommendation seeks to predict a user's future location based on their past mobility patterns, a task essential for personalized location-based services. To enhance semantic modeling in this context, recent studies have applied large language models (LLMs), where early approaches relied on randomly assigned POI identifiers with limited representational capacity. More recent work has introduced semantic identifiers (SIDs) to better capture spatial and contextual correlations, leading to improved prediction accuracy. However, these approaches face several critical challenges: a semantic gap between LLM's language semantics and POI-specific collaborative semantics, and the generation of invalid SIDs that do not correspond to real POIs. To tackle these challenges, we propose a novel framework called BLCSRec for bridging language and collaborative semantics in next POI recommendation. Specifically, we introduce an LLM-based POI profile generation method that summarizes user trajectories and integrates POI attributes with visitor information, and further employ an RQ-VAE to encode addresses, category, and these enriched textual profiles into semantic identifiers that capture both static attributes and collaborative context. To alleviate the gap between language and collaborative semantic, we incorporate explicit alignment tasks that map SIDs to/from textual descriptions and implicit alignment tasks that predict next POIs in asymmetric semantic formats. Furthermore, we employ GRPO reinforcement learning with a hierarchical reward structure to suppress invalid SID generation and enhance accuracy. Extensive experiments on three public datasets (NYC, TKY, and CA) demonstrate that our method consistently outperforms strong LLM-based baselines, achieving improvements of 7.3% in Acc@1 on NYC and 3.5% on CA, along with substantial reductions in invalid SID generation.

## 1 Introduction

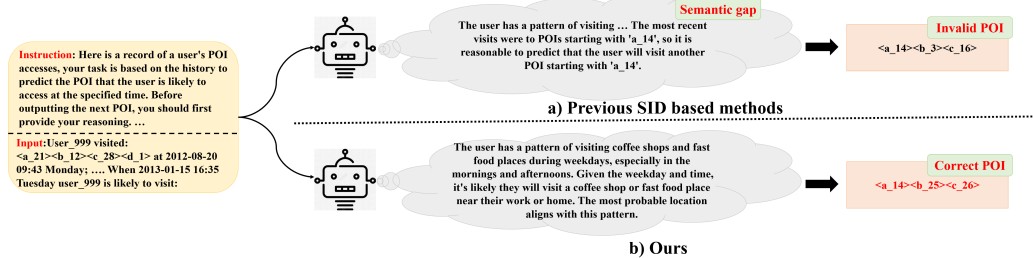

Figure 1: Chanllenge and Case Study

The proliferation of Location-Based Social Networks (LBSNs) has generated an unprecedented volume of user mobility data, typically captured as chronological sequences of check-ins at various Points-of-Interest (POIs). Capitalizing on these rich datasets, the task of next POI recommendation(Zhang et al., 2025a; Yu et al., 2024) has emerged as a pivotal research direction. This task is

fundamentally concerned with predicting a user's subsequent destination by modeling the intricate patterns embedded within their historical movement trajectories.

Early approaches(Feng et al., 2015; Graves, 2012; Wu et al., 2022; Luo et al., 2021; Yang et al., 2022; Yan et al., 2023; Luo et al., 2023; Feng et al., 2024) to next POI recommendation have predominantly utilized classic deep learning models, such as Long Short-Term Memory (LSTM) networks(Hochreiter & Schmidhuber, 1997) and Graph Neural Networks (GNNs)(Scarselli et al., 2009), to capture users' temporal or spatial preferences. Following the demonstrated success of large language models (LLMs) in various recommendation tasks (Li et al., 2024b), recent studies have extended their use to next POI recommendation. While some approaches(Li et al., 2024a) construct prompts using randomly assigned POI identifiers, this setting limits the model's ability to capture semantic relations among locations. In contrast, prior work in other recommendation domains has demonstrated that using semantic identifiers (SIDs), where items with similar characteristics are assigned more similar SIDs, allows LLMs to better capture the relationships and similarities among items(Zhou et al., 2025; Zheng et al., 2024a; Rajput et al., 2023). Motivated by these findings, several recent efforts(Wang et al., 2025) have introduced SIDs into next POI recommendation, enabling LLMs to better exploit spatial and contextual correlations and thus improving predictive accuracy.

Despite their promising performance in next POI recommendation, approaches utilizing LLMs with SIDs are constrained by two primary limitations, as shown in Figure 1. a) **Semantic gap**: there exists a large gap between the language semantics modeled by LLMs and collaborative semantics implied by recommender systems. LLMs process natural language through token IDs, whereas recommendation systems utilize POI-specific identifiers (in this case, SIDs). This fundamental misalignment hinders the ability to fully harness the LLM's representational power for recommendation tasks, as the models effectively learn within disparate semantic spaces. As shown in Figure 1a), when generating SIDs, the LLM fails to capture the semantics of index tokens and instead infers user preferences solely from their visit frequency. b) **Invalid POIs**: the inherent constraints of LLMs, specifically their finite context windows and imperfect instruction adherence, render it infeasible to inform the model of all valid SIDs through enumeration. Consequently, during unconstrained generation, the LLM is prone to producing spurious SIDs that do not correspond to any actual POI.

To address these challenges, we propose Bridging Language and Collaborative Semantics for Next POI Recommendation (BLCSRec). The framework first leverages an LLM to construct user profiles that capture individual preferences from check-in records, and POI profiles that summarize visiting patterns based on user preferences and temporal context. These profiles enrich the semantic space for SID generation, thereby enhancing the discriminability of POI representations and also provide LLM-comprehensible inputs and outputs, paving the way for an explicit alignment task. Each POI is then assigned a SID by RQ-VAE(Lee et al., 2022) using its address, category, and profile. The sequence of SIDs derived from a user's visited POIs forms the training data for the LLM-based Sequential POI Prediction task. To mitigate the semantic gap, BLCSRec introduces two complementary strategies: (i) Explicit Alignment, which requires the LLM to map between SIDs and textual descriptions during training, grounding tokens in semantic meaning; and (ii) Implicit Alignment, which predicts the next POI's SID or text from sequences of check-in records, encouraging the model to exploit SID–semantic relations in sequential prediction. The three datasets are jointly used to train the LLM. Moreover, since reinforcement learning has been shown to increase the likelihood of generating preference-consistent samples while reducing the probability of non-preferred outputs(Yue et al., 2025; Xiong et al., 2025), we design a hierarchical reward structure that assigns higher rewards to correct SIDs, lower rewards to valid but incorrect SIDs, and the lowest rewards to invalid SIDs. GRPO(Shao et al., 2024) is then employed to suppress invalid generations and improve overall prediction accuracy. Our main contributions can be summarized as follows:

- We find that existing SID-based approaches employing LLMs for next POI recommendation mainly suffer from two issues: a semantic gap between tokenized representations and POI-specific identifiers, and a non-negligible risk of generating invalid SIDs.

- We construct POI profiles to support SID generation and introduce both explicit and implicit alignment tasks, which alleviate the semantic gap by linking SIDs with their semantic representations. In addition, GRPO is incorporated to reduce the probability of producing invalid SIDs and to further improve prediction accuracy.

- We evaluate BLCSRec against strong baselines on three benchmark datasets (NYC, TKY, and CA), achieving consistent improvements in Acc@1 (e.g., +7.3% on NYC and +3.5%

on CA over the best baseline) while substantially reducing the generation of invalid SIDs, which demonstrates the effectiveness of our method.

## 2 RELATED WORK

Early efforts in next POI recommendation typically formulated the problem as a sequential prediction task, often relying on handcrafted models such as Markov chains(He et al., 2016) or factorization techniques(Feng et al., 2015). With the advancement of deep learning, recurrent neural networks(RNNs) became the mainstream, where variants were designed to capture spatial-temporal dynamics and model both short- and long-term user preferences(Graves, 2012; Wu et al., 2022). More recently, attention-based architectures have shown superior performance by providing greater flexibility in modeling dependencies. Transformer-based models(Luo et al., 2021; Yang et al., 2022; Luo et al., 2023; Feng et al., 2024), such as GETNext(Yang et al., 2022), integrates a global trajectory flow graph with Transformer to jointly model collaborative, spatial-temporal, and semantic contexts. Parallel to this line of work, graph neural networks(GNNs) have been adopted to capture user–POI relationships beyond individual sequences(Zhang et al., 2021; Yan et al., 2023).

Following the demonstrated superior performance of LLMs in various recommendation tasks(Li et al., 2024b), recent research(Wang et al., 2023; Liu et al., 2024; Li et al., 2024a; Wang et al., 2025) has begun to explore their application to next POI recommendation. LLM4POI(Li et al., 2024a) involves constructing a prompt from a user's historical sequence of visited POI identifiers and tasking the LLM with generating the ID for the next recommended POI. Recognizing that many high-performing LLM-based recommenders rely on semantic IDs (SIDs)(Zhou et al., 2025; Zheng et al., 2024a; Rajput et al., 2023), GNPR-SID(Wang et al., 2025) introduces SIDs by assigning a unique semantic identifier to each POI, ensuring that similar POIs possess more proximate SID representations. By constructing the input sequence with these SIDs, the LLM can more effectively capture the latent correlations between POIs, including spatial proximity and other contextual attributes, ultimately leading to improved recommendation accuracy.

Although existing LLM-based methods leveraging SIDs have achieved state-of-the-art performance in next POI recommendation, they overlook two critical issues. First, there remains a semantic gap between the language-level semantics captured by LLMs and the collaborative semantics inherent in recommender systems. Second, LLMs can occasionally generate invalid SIDs that do not correspond to any real POI, which may degrade recommendation reliability. To address these issues, we introduce a set of alignment tasks to mitigate the semantic gap and employ GRPO(Shao et al., 2024) to reduce the likelihood of generating invalid SIDs.

## 3 METHOD

To address the two primary challenges in existing SID-based next POI recommendation approaches, we propose tailored solutions. For the semantic gap, we first derive user preferences and temporal patterns by analyzing historical visitation records and timestamps for both users and POIs, then introduce a series of supplementary semantic alignment tasks during the LLM's Supervised Fine-Tuning (SFT) process. These tasks encompass explicit alignment, which directs the LLM to generate corresponding SIDs/textual descriptions from POI textual descriptions/SIDs, and implicit alignment, which involves predicting subsequent POI SIDs or textual descriptions given input sequences with textual descriptions or SIDs. For invalid SID generation, we implement an additional Reinforcement Learning phase after SFT using the GRPO training methodology to encourage the model to output accurate and existing POI SIDs.

Our methodology comprises three primary phases: semantic ID construction, language-collaborative POI SID alignment, and reinforcement learning, as illustrated in Figure 2.

### 3.1 SEMANTIC ID CONSTRUCTION

Similar to approaches(Li et al., 2025; Deng et al., 2025; Zheng et al., 2024b; Yang et al., 2025b) in previous recommender systems, we eschew raw POI IDs in favor of vector quantization techniques, representing each POI with a compact set of discrete indices. While these indices are traditionally constructed using textual information, POIs in next-POI recommendation tasks typically have

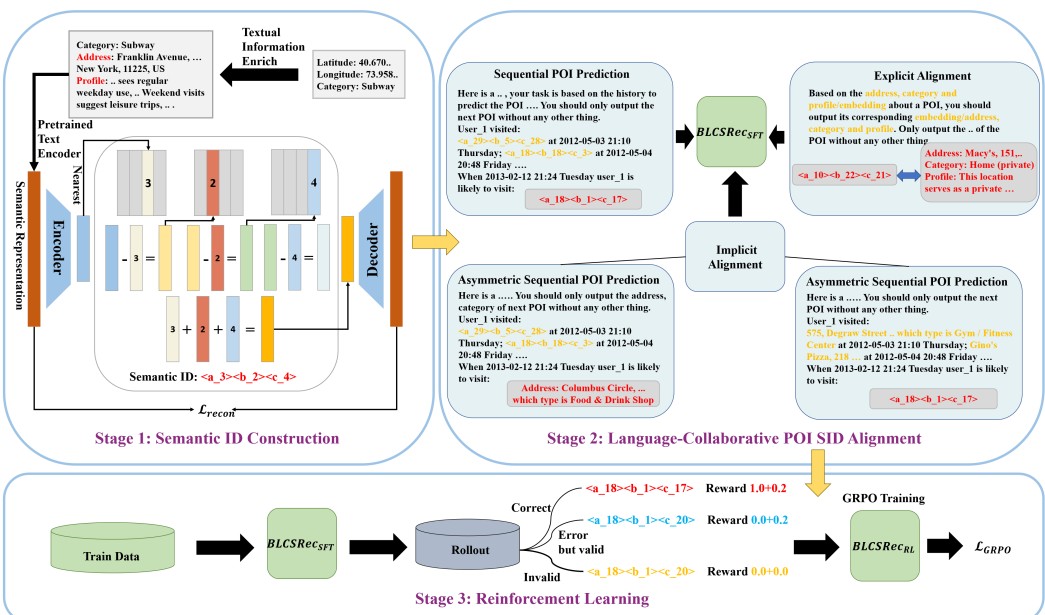

Figure 2: The overall architecture of our proposed framework.

limited text data, often consisting only of geographical coordinates and category information. To generate distinctive and semantically rich semantic IDs for each POI, we first construct semantic representation for each POI and subsequently employed RQ-VAE(Lee et al., 2022) for semantic ID construction.

### 3.1.1 SEMANTIC REPRESENTATION CONSTRUCTION

To construct semantically rich text representations, we enhance each POI's textual description beyond its existing categorical information. Specifically, we enrich the spatial features and developed comprehensive POI profiles that capture both collaborative and temporal patterns.

**Spatial Features**  For next-POI recommendation tasks, the conventional use of precise latitude-longitude coordinates poses challenges for LLMs due to their limitation in processing high-precision numerical inputs(Yang et al., 2025a). Although prior work(Wang et al., 2025) addressed this through Plus Code[1] discretization, this approach suffers from location ambiguity within the same code region. To solve these problems, We proposes converting coordinate pairs into U.S.-style address strings (number + street + city + state) via standard mapping services[2]. This representation maintains spatial granularity through shared address components among proximate locations while ensuring distinctiveness for POIs, thereby enabling LLMs to effectively capture spatial relationships without relying on precise numerical values.

**POI Profiles**  Inspired by previous approaches(Liu et al., 2025) that generate item profiles from interaction patterns when basic item information is limited, we develop a two-stage LLM-based profiling process. First, we generate comprehensive user profiles capturing travel preferences by feeding each user's historical POI interaction sequences (including locations, categories, and timestamps) into an LLM to summarize their categorical preferences, temporal patterns, and geographical trends. Subsequently, for each POI, we enrich its textual information by prompting the LLM to synthesize its collaborative and temporal patterns based on its location, category, and visitor information (including user profiles and visit timestamps). This approach enables the construction of semantically-rich textual representations that effectively capture both explicit and implicit POI characteristics.

---

[1]https://maps.google.com/pluscodes/

[2]https://nominatim.org/

Finally, we concatenate each POI's categorical information, spatial features, and generated profile into a comprehensive textual description $text$, which serves as input to a pre-trained text encoder $E_t$ to obtain the POI's semantic representation $p_s$.

### 3.1.2 POI QUANTIZATION

After obtaining the POI semantic representations, we employ vector quantization to generate discrete SID for each POI. Specifically, we utilize a RQ-VAE(Lee et al., 2022) with the semantic representations as input to generate POI semantic IDs $SID$. RQ-VAE implements a multi-stage vector quantization process that recursively quantizes residual vectors from coarse to fine levels to produce a set of index tokens. For each semantic representation $p_s$, RQ-VAE first encodes it into a latent representation $z$. At each level $l$, we maintain a codebook $\mathcal{C}^l = \{v_k^l\}_{k=1}^K$, where each codebook vector $v_k^l$ represents a learnable cluster center and $K$ is the number of cluster centers. The residual quantization process can be formulated as:

$$c_i = \arg\min_k \|e_i - v_k^i\| \tag{1}$$

$$e_{i+1} = e_i - v_{c_i}^i \tag{2}$$

$$SID = [c_1, c_2, ..c_L] \tag{3}$$

where $c_i$ is the i-th index token of the semantic ID, $L$ denotes the number of quantization levels, and $e_i$ is the residual vector in the i-th RQ level, and we set $e_1 = z$.

After quantizing the semantic representation into a SID, we obtain the corresponding latent representation $\hat{z}$ by dequantizing each level's index using the associated codebook vectors according to $\hat{z} = \sum_{i=1}^L v_{c_i}^i$. This latent representation is then passed through a decoder to reconstruct the semantic representation $\hat{p_s}$. The overall loss function of RQ-VAE is defined as follows.

$$\mathcal{L}_{recon} = \|p_s - \hat{p_s}\|_2^2 \tag{4}$$

$$\mathcal{L}_{RQ} = \sum_{i=1}^L \|sg[e_i] - v_{c_i}^i\|_2^2 + \beta\|e_i - sg[v_{c_i}^i]\|_2^2 \tag{5}$$

$$\mathcal{L}_{RQ-VAE} = \mathcal{L}_{recon} + \mathcal{L}_{RQ} \tag{6}$$

The loss function consists of two components: $L_{\text{recon}}$, which measures the reconstruction error, and $L_{\text{RQ}}$, which encourages the residual vectors to align closely with their assigned codebook entries. The stop-gradient operator sg[·] is applied to prevent gradient flow through discrete variables, and the coefficient $\beta$, typically set to 0.25.

It is worth noting that the trained RQ-VAE may produce a non-negligible rate of conflicts, where multiple POIs are assigned the same SID. To ensure that each POI is associated with a distinct identifier, we append a unique suffix to the shared semantic ID. For instance, if two POIs are mapped to the same ID sequence <a_1><b_1><c_1>, they are further distinguished as <a_1><b_1><c_1><d_0> and <a_1><b_1><c_1><d_1>, respectively.

### 3.2 LANGUAGE-COLLABORATIVE POI SID ALIGNMENT

Although the constructed SIDs allow POIs with similar meanings to share similar prefix index tokens, which improves their accessibility to LLMs, the models themselves lack awareness of the semantic content encoded in index tokens such as <a_1>. To bridge this gap, we design a series of semantic alignment tasks to incorporate both language and collaborative semantics into the LLMs through fine-tuning. These tasks include: (1) sequential POI prediction, (2) explicit alignment, and (3) implicit alignment.

### 3.2.1 SEQUENTIAL POI PREDICTION

Our method leverages LLM for next-POI prediction as the primary training objective. Specifically, we construct personalized recommendation instruction based on users' historical check-in patterns. These instruction, along with check-in histories, serve as input to the LLM to predict the POI a

target user will likely visit at a specified future time. The user's check-in history is represented as a chronologically ordered sequence of POI semantic IDs.

Notably, the temporal aspects of historical check-in records are significant in two distinct dimensions: the timestamp information captures user-specific behavioral patterns at a granular level, while the day-of-week features encode cyclic behavioral variations in weekly routines (exemplified by students' distinct patterns of visiting academic institutions on weekdays versus preferring residential or recreational venues on weekends). Consequently, we concatenate the chronologically ordered POI SIDs with their corresponding check-in timestamps and day-of-week indicators. This integrated approach enables more comprehensive modeling of personalized user information, effectively synthesizing collaborative semantics across both spatial and temporal dimensions.

However, merely tuning LLM through this approach fails to ensure adequate perception of feature information embedded within each index token of the SIDs. To address this limitation, we incorporate both explicit and implicit alignment tasks to bridge the semantic gap between language and collaborative semantics in LLMs.

### 3.2.2 EXPLICIT ALIGNMENT

While our POI SIDs are constructed from POI addresses, categories, and profiles, they establish only weak correlations among POIs with similar linguistic semantics through shared prefix index tokens. This results in LLM's limited ability to learn semantic information from these weak associations. To enhance LLM's semantic understanding of each index token, we propose two explicit index token-language alignment tasks for LLM tuning.

Our alignment objectives are twofold: On the one hand, we aim to enable LLM to interpret a POI's address, category, and profile when presented with its semantic ID. On the other hand, we seek to develop LLM's capability to construct the appropriate semantic ID when given a POI's address, category, and profile information. We implement this bidirectional alignment through two sequential tasks: first instructing LLM to generate the corresponding address, category, and profile from a POI's semantic ID, then directing it to reconstruct the semantic ID based on these attributes.

### 3.2.3 IMPLICIT ALIGNMENT

While explicit alignment tasks enable LLM to perceive the mapping relationship between POI textual information and semantic IDs, this understanding alone may be insufficient for practical application, analogous to how memorizing mathematical formulas doesn't guarantee problem-solving ability(Shao et al., 2024). In the recommendation context, LLM might struggle to effectively utilize these learned mappings during actual recommendation tasks. Therefore, we designed two asymmetric POI prediction tasks as implicit alignment tasks specifically tailored to recommendation scenarios.

In sequential item prediction, both check-in histories and target POI are formatted using semantic ID representations, constituting a symmetric task where both conditions and targets are semantically based. To enhance semantic alignment further, we increase prediction complexity by varying the forms of conditions and targets, yielding different combinations of POI semantic representations. Specifically, we implement two asymmetric representations: First, replacing target POI semantic IDs with corresponding addresses and categories, directing LLM to generate POI attributes from semantic ID sequences; Second, substituting check-in history's POI semantic IDs with addresses and categories, guiding LLM to generate expected POI semantic ID.

### 3.2.4 SUPERVISED FINE-TUNING

Following dataset construction for all three tasks, we integrated them for LLM's SFT. Following conventional SFT practices for decoder-only architectures, we optimize the model parameters through token-level cross-entropy loss, maximizing the conditional probability of subsequent tokens given their preceding context, thereby obtaining our fine-tuned model **BLCSRec$_{\textbf{SFT}}$**.

Table 1: Comparison of different models on three datasets: NYC, TKY, and CA. We present the model inputs including the POI representation approach (POI Repre.) and whether visit timestamps are utilized.

| Model | Inputs | | $Acc@1$ | | | $rate_i$ | | |
|-------|--------|-----|-----|-----|-----|-----|-----|-----|
| | POI Repre. | Seq | NYC | TKY | CA | NYC | TKY | CA |
| PRME | RID | - | 0.1159 | 0.1052 | 0.0521 | - | - | - |
| LSTM | RID | - | 0.1305 | 0.1335 | 0.0665 | - | - | - |
| PLSPL | RID | - | 0.1917 | 0.1889 | 0.1072 | - | - | - |
| STAN | RID | - | 0.2231 | 0.1963 | 0.1104 | - | - | - |
| GETNext | RID | - | 0.2435 | 0.1829 | 0.1357 | - | - | - |
| STHGCN | RID | - | 0.2734 | 0.2950 | 0.1730 | - | - | - |
| TPG | RID | - | 0.2555 | 0.1420 | 0.1749 | - | - | - |
| ROTAN | RID | - | 0.3106 | 0.2458 | 0.2199 | - | - | - |
| LLM4POI | RID | 300 | 0.3372 | 0.3035 | 0.2065 | - | - | - |
| GNPR-SID | SID | 50 | 0.3310 | 0.2801 | 0.2119 | 1.42% | 1.31% | 1.63% |
| **BLCSRec$_{\mathbf{SFT}}$** | SID | 50 | 0.3589 | 0.2979 | 0.2239 | 0.72% | 0.40% | 1.40% |
| **BLCSRec$_{\mathbf{RL}}$** | SID | 50 | **0.3618** | **0.3049** | **0.2277** | **0.00%** | **0.05%** | **1.12%** |

### 3.3 Reinforcement Learning

As GRPO has been demonstrated to steer model outputs toward higher-reward outcomes(Yue et al., 2025), in the reinforcement learning stage, we utilize the GRPO algorithm to enhance answer accuracy and reduce the likelihood of the model generating invalid SIDs.

In our RL stage, we incorporate two reward signals: the validity reward and the accuracy reward. The validity reward encourages the model to generate semantic IDs corresponding to real-world POIs; a score of 0.2 is assigned if the predicted semantic ID maps to an existing POI, and 0 otherwise. The accuracy reward promotes precise prediction by assigning a score of 1 when the predicted semantic ID $\hat{y}_i$ matches the ground truth $y_i$, and 0 otherwise.

## 4 Experiments

### 4.1 Experimental Settings

**Datasets** We evaluate our approach on three real-world datasets: Foursquare NYC(Yang et al., 2015), Foursquare TKY(Yang et al., 2015), and Gowalla CA(Cho et al., 2011). Following previous works(Yan et al., 2023), we preprocess each dataset by filtering out POIs and users with fewer than 10 check-ins. The data is chronologically sorted, with an 8-1-1 split for training, validation, and testing. To maintain consistency between training and evaluation, we exclude users and POIs from the test set that do not appear in the training data. Check-in records are then grouped by users and arranged chronologically. During evaluation, each user's last visited POI serves as ground truth, while previous visits constitute the input sequence. Notably, during training, users' historical records are treated as sequential input and concatenated with the test set to a specified length before inference.

**Evaluation Metrics** Consistent with previous LLM-based methods(Li et al., 2024a; Wang et al., 2025), our model generates a single POI recommendation per inference. Therefore, we adopt the accuracy metric for top-1 recommendation $acc@1$. Additionally, for LLM-based methods utilizing semantic IDs as input and output, we introduce an invalid SID rate metric $rate_i$ to assess the probability of generating invalid SIDs.

### 4.2 Main Results

We evaluate our approach against current state-of-the-art methods across the NYC, TKY, and CA datasets, with results presented in Table 1. Our method **BLCSRec$_{\mathbf{RL}}$** demonstrates significant

improvements in $Acc@1$ metrics, achieving gains of 7.3%, 0.5%, and 3.5% respectively over the best existing approaches. Notably, even our SFT model **BLCSRec$_{SFT}$** surpasses current state-of-the-art performance on NYC and CA datasets. On the TKY dataset, our method achieves comparable performance to LLM4POI(Li et al., 2024a) using only 50 check-in records, versus LLM4POI's 300 records. Compared to method GNPR-SID(Wang et al., 2025), which also employs semantic IDs, both our SFT and RL models exhibit superior performance in both accuracy and invalid rate metrics, demonstrating enhanced effectiveness and robustness. We attribute these improvements to three key factors:

- Enhanced POI Representation: Our approach generates comprehensive POI profiles incorporating visit preferences and temporal characteristics, enriching textual information and enabling more distinctive semantic IDs with reduced conflicts.

- Semantic Alignment: The introduction of alignment tasks during SFT bridges the semantic gap between natural language and semantic IDs, enabling the LLM to better capture the semantic content of each index token, simultaneously improving accuracy while reducing invalid SIDs occurrences.

- RL Optimization: The post-SFT reinforcement learning phase effectively increases the probability of generating correct semantic IDs while reducing invalid SIDs predictions.

## 4.3 ABLATION STUDY

The ablation study was conducted on the NYC and CA datasets to examine the effect of each component in our framework. Evaluation was based on $Acc@1$ and $rate_i$. Beginning with the pretrained Qwen2.5-7B-Instruct(Yang et al., 2024) model without POI profiles, we incrementally added POI profile incorporation(profile), sequential POI prediction fine-tuning(SP), explicit(EA) and implicit alignment(IA) objectives, and finally GRPO training(GRPO).

The results in Table 2 show steady gains in accuracy and reductions in invalid SIDs generation. On NYC, $Acc@1$ improves from 0.2601 (baseline) to 0.3618 with GRPO, while $rate_i$ drops from 4.35% to 0.00%. A similar trend is observed on CA, where $Acc@1$ rises from 0.1695 to 0.2277 and $rate_i$ decreases from 1.85% to 1.12%. These findings indicate that each component contributes to accuracy improvement, with explicit and implicit alignment yielding notable gains, and GRPO providing the most effective balance between accuracy and invalid SIDs reduction.

| Method | NYC | | CA | |
|---|---|---|---|---|
| | $Acc@1$ | $rate_i$ | $Acc@1$ | $rate_i$ |
| Qwen2.5-7B | 0.2601 | 4.35% | 0.1695 | 1.85% |
| + profile | 0.2901 | 4.02% | 0.1718 | 1.72% |
| + SP | 0.3357 | 1.30% | 0.2151 | 1.62% |
| + EA | 0.3546 | 1.12% | 0.2179 | 1.59% |
| + IA | 0.3589 | 0.72% | 0.2239 | 1.41% |
| + GRPO | 0.3618 | 0.00% | 0.2277 | 1.12% |

Table 2: Ablation Study.

## 4.4 SEMANTIC ANALYSIS

To evaluate whether our alignment task effectively language and collaborative semantics in LLM, we conducted a semantic analysis on the NYC dataset from two perspectives: LLM token embedding visualization analysis and LLM performance on semantically similar negative items.

### 4.4.1 EMBEDDING VISUALIZATION ANALYSIS

We investigate the relationship between POI index tokens and the original semantic space of the LLM. Following prior studies(Zheng et al., 2024b), we apply Principal Component Analysis (PCA)(Yang et al., 2004) to visualize embeddings of different token types. In Figure 3a), "Tokens w/ Alignment" refers to all SID index tokens produced by the LLM when the alignment task is incorporated, whereas "Tokens w/o Alignment" denotes all SID index tokens derived from the LLM trained without the alignment task during SFT, and "POI Texts" refers to tokens derived from POI addresses, categories and profiles. The 2D visualization indicates that, after incorporating the alignment task, the distribution of SID index tokens in the semantic space becomes closer to the

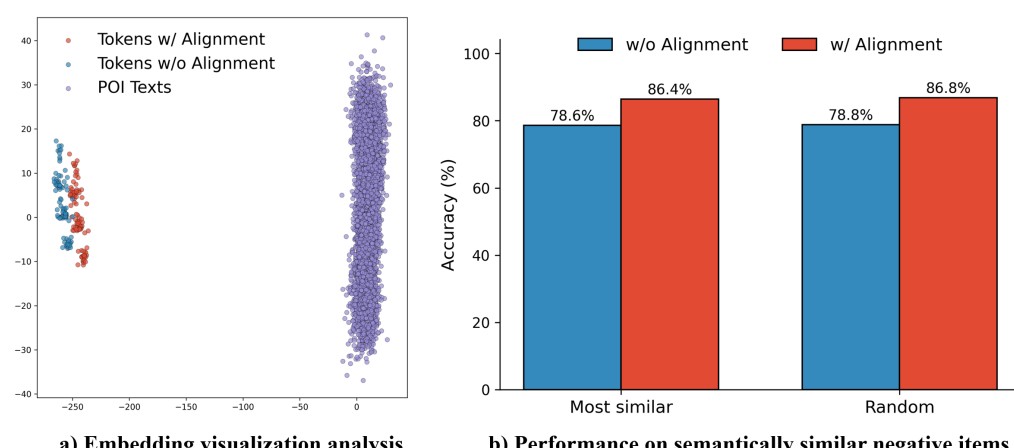

a) Embedding visualization analysis    b) Performance on semantically similar negative items

Figure 3: Semantic Analysis

LLM semantic space of POI textual information compared to the setting without alignment. This observation indicates that the alignment task can effectively reduce the semantic gap between LLM language and collaborative semantics.

### 4.4.2 PERFORMANCE ON SEMANTICALLY SIMILAR NEGATIVE ITEMS

To examine why combining language and collaborative semantics benefits LLMs in next POI recommendation, we design an auxiliary ranking task with negative samples. Specifically, two types of negatives are constructed: (1) most similar negatives, identified by cosine similarity between POI text embeddings, and (2) random negatives, sampled uniformly from the valid POI set. The evaluation reuses the same test set as the sequential POI prediction task, where the model must distinguish the ground-truth item from a negative, the performance is reported in terms of accuracy.

The results showed in Figure 3b) indicate that, after incorporating the alignment tasks, the LLM outperforms its non-aligned counterpart in both distinguishing POIs with similar language semantics and identifying random negative POIs. This demonstrates that our proposed explicit and implicit alignment tasks effectively reduce the semantic gap between recommendation and natural language tasks, enabling the model to capture semantic information within index tokens and thereby improve recommendation performance.

## 5 CONCLUSION

In this work, we address two critical issues in LLM-based next POI recommendation systems related to SID integration: the semantic gap between language semantics modeled by LLMs and collaborative semantics in POI identifiers, and the generation of invalid SIDs that undermine prediction reliability. To tackle these challenges, we propose BLCSRec, which leverages LLMs to construct user and POI profiles capturing preferences, temporal patterns, and collaborative signals, thereby enriching SID generation via RQ-VAE for more discriminative representations. Furthermore, we introduce explicit alignment tasks to ground SIDs in textual semantics through bidirectional mapping, and implicit alignment tasks to encourage the model to exploit SID-semantic relations in asymmetric sequential prediction. Additionally, we employ GRPO reinforcement learning with a hierarchical reward mechanism that prioritizes correct SIDs, penalizes invalid ones. Experimental results on three public datasets (NYC, TKY, and CA) show that our approach consistently outperforms strong baselines, with improvements of 7.3% in Acc@1 on NYC and 3.5% on CA, while substantially reducing invalid SID generation. These findings underscore the importance of semantic alignment and reinforcement strategies for enhancing the robustness and accuracy of LLM-based recommender systems, offering insights for broader applications in sequential recommendation tasks involving spatial-temporal data.

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

# A    APPENDIX

## A.1    GENAI USAGE DISCLOSURE

LLMs play a central role in our work. First, we employ LLMs to generate profiles for users and POIs. Second, since our approach follows a generative recommendation paradigm, the underlying generative model we adopt is also an LLM. In addition, we utilize LLMs to refine the writing of this paper. Beyond these uses, no generative models are involved in any other part of our work, including the codebase, figures, tables, or experimental data.

## A.2    BASELINES

Our experimental evaluation encompasses diverse baseline approaches across five categories: Traditional Sequential Models including PRME(Feng et al., 2015), a metric embedding approach for personalized ranking, LSTM(Graves, 2012), a recurrent neural architecture for sequence modeling, and PLSPL(Wu et al., 2022), a personalized sequential recommendation framework; Transformer-based Architectures comprising STAN(Luo et al., 2021), which incorporates spatio-temporal attention mechanisms, and GETNext(Yang et al., 2022), leveraging transformer encoders for POI prediction; Temporal-aware Frameworks including TPG(Luo et al., 2023), employing timestamp-guided prediction strategies, and ROTAN(Feng et al., 2024), integrating temporal awareness in POI recommendations; Graph-based Approaches represented by STHGCN(Yan et al., 2023), utilizing hierarchical graph convolutions for spatio-temporal modeling; and LLM-based Methods: LLM4POI(Li et al., 2024a), the current state-of-the-art approach that utilizes random numeric identifiers and comprehensive user check-in histories; GNPR-SID(Wang et al., 2025), which leverages semantic IDs constructed from discretized POI features and processes user check-in sequences through these semantic representations.

## A.3    IMPLEMENTATION DETAILS

During semantic ID construction, we employ GPT-41-mini[3] for generating user and POI profiles, followed by Qwen3-Embedding-0.6B(Zhang et al., 2025b) for text representation generation. The RQ-VAE(Lee et al., 2022) module incorporates 3 codebook layers, each comprising 32 index tokens for NYC and 64 for TKY and CA datasets, with a vector dimensionality of 64. For the base model, we select Qwen2.5-7B-Instruct(Yang et al., 2024), maintaining comparable parameter scale with previous works(Li et al., 2024a; Wang et al., 2025). During the SFT stage, we apply LoRA(Hu et al., 2022) fine-tuning to all linear layers, with dataset-specific rank settings of 128 for NYC and 1024 for both TKY and CA. Training is conducted using the LLaMA-Factory(Zheng et al., 2024c) framework on 4 RTX 4090 GPUs, with a constant learning rate of 1e-5, a batch size of 1 per GPU, gradient accumulation steps set to 4, and a sequence length of 4096 tokens. In the RL stage, we fine-tune the linear layers within the LLM's Attention modules using LoRA, employing a learning rate of 1e-6 and lora rank of 128. The training data used for the sequential POI prediction task is reused as training data for GRPO. We initialize GRPO training from model **BLCSRec$_{\mathbf{SFT}}$**. This stage is implemented using the ms-swift(Zhao et al., 2024) framework across 8 RTX 4090 GPUs, with a batch size of 1 per GPU, gradient accumulation steps set to 4, $G$ set to 16 and a sequence length of 4096 tokens. To address potential convergence issues arising from the underrepresentation of our primary sequential POI prediction task, we augmented its presence by triplicating the corresponding data. To facilitate the comprehension of LLMs, we first translate the Japanese addresses in the TKY dataset, obtained from mapping services, into English using translation software[4] before further utilization.

## A.4    TRAINING DATA EXAMPLES

This section illustrates the training samples for the three tasks in our Language-Collaborative POI SID Alignment stage. An example for the sequential POI prediction task is presented as follows:

---

[3]https://openai.com/index/gpt-4-1/

[4]https://translate.google.com

**Instruction**: Here is a record of a user's POI accesses, your task is based on the history to predict the POI that the user is likely to access at the specified time. You should only output the next POI without any other thing.
**Input**: User_3 visited: <a_6><b_7><c_14> at 2012-04-07 23:09 Saturday; <a_24><b_16><c_31><d_2> at 2012-04-08 02:13 Sunday; .... When 2012-09-15 16:58 Saturday user_3 is likely to visit:
**Output**: <a_5><b_0><c_20>

The training data for explicit alignment task is demonstrated as follows:

**Instruction**: Based on the embedding about a POI, you should output its corresponding address, category and profile. Only output the address, category and profile of the POI without any other thing.
**Input**: <a_5><b_0><c_20>
**Output**: Address: John F. Kennedy International Airport, ...
Category: Airport
Profile: ...

- - - - - - - - - - - - - - - - - - - - - - - - - - - - - - - - - - - - - - - - - - - - - - - - - -

**Instruction**: Based on the address, category and profile about a POI, you should output its corresponding embedding. Only output the embedding of the POI without any other thing.
**Input**: Address: John F. Kennedy International Airport, ...
Category: Airport
Profile: ...
**Output**: <a_5><b_0><c_20>

The training data for implicit alignment task is demonstrated as follows:

**Instruction**: Here is a record of a user's POI accesses, your task is based on the history to predict the POI that the user is likely to access at the specified time. You should only output the address, category of next POI without any other thing
**Input**: User_3 visited: <a_6><b_7><c_14> at 2012-04-07 23:09 Saturday; <a_24><b_16><c_31><d_2> at 2012-04-08 02:13 Sunday; .... When 2012-09-15 16:58 Saturday user_3 is likely to visit:
**Output**: John F. Kennedy International Airport, JFK Access Road, Queens, Queens County, City of New York, New York, 11430, United States which type is Airport

- - - - - - - - - - - - - - - - - - - - - - - - - - - - - - - - - - - - - - - - - - - - - - - - - -

**Instruction**: Here is a record of a user's POI accesses, your task is based on the history to predict the POI that the user is likely to access at the specified time. You should only output the next POI without any other thing.
**Input**: User_3 visited: 115, Court Street, ... which type is Movie Theater at 2012-04-07 23:09 Wednesday; Shake Shack, 409, Fulton Mall, ... which type is Burger Joint at 2012-04-08 02:13 Wednesday; .... When 2012-09-15 16:58 Sunday user_3 is likely to visit:
**Output**: <a_3><b_7><c_44>

## A.5 GRPO

During the training of GRPO, for each training iteration, given the input $q$, we sample $G$ candidate outputs from the previous policy $\pi_{\text{old}}$. Each candidate receives a reward $r$, and we compute the group-relative advantage $\hat{A}_i$ for candidate $i$ as

$$\hat{A}_i = \frac{r_i - \mu}{\sigma} \tag{7}$$

where $\mu$ and $\sigma$ denote the mean and standard deviation of rewards within the candidate group. Outputs with above-average rewards thus gain higher advantages, guiding prioritized optimization. The policy update then maximizes the following clipped surrogate objective:

$$\mathcal{L}_{GRPO}(\theta) = \mathbb{E}_{o_i \sim \pi_\theta} \left[ \frac{1}{G} \sum_{i=1}^{G} \min \left( \rho_i \hat{A}_i, \ \text{clip}(\rho_i, 1-\epsilon, 1+\epsilon) \hat{A}_i \right) \right] - \beta \, \text{KL} \left[ \pi_\theta \, \| \, \pi_{\text{old}} \right], \quad (8)$$

where $\pi_\theta$ is the new policy, $\rho_i = \frac{\pi_\theta(o_i|q)}{\pi_{\text{old}}(o_i|q)}$ is the importance-sampling ratio, $\epsilon$ is the clipping hyperparameter, and the KL divergence term with weight $\beta$ constrains policy deviation for stability which is calculated as $\text{KL}\left[ \pi_\theta \, \| \, \pi_{\text{old}} \right] = \frac{\pi_{\text{old}}}{\pi_\theta} - log \frac{\pi_{\text{old}}}{\pi_\theta} - 1$.

### A.6 SUPERVISED FINE-TUNING LOSS

The SFT loss is as follows:

$$\mathcal{L}_{sft} = \sum_{t=1}^{T} p_\theta(x_t | x_1, x_2, ... x_{t-1}) \quad (9)$$

Where $T$ denotes the length of the output sequence, $x_t$ represents the token being predicted, $x_1, x_2, ..., x_{t-1}$ refers to the preceding tokens in the sequence, $\theta$ denotes the trainable model parameters, and $p_\theta(x_t|x_1, x_2, ...x_{t-1})$ represents the predicted probability distribution of the model.

### A.7 EVALUATION METRICS DEFINITION

$acc@1$ is defined as:

$$acc@1 = \frac{1}{N} \sum_{i=1}^{N} \mathbb{I}(\hat{y}_i = y_i) \quad (10)$$

where $N$ denotes the number of test samples, $\mathbb{I}(\cdot)$ represents the indicator function that yields 1 when the condition is satisfied and 0 otherwise, and $\hat{y}_i, y_i$ denote the model's prediction and ground truth, respectively. And $rate_i$ is formulated as:

$$rate_i = \frac{1}{N} \sum_{i=1}^{N} \mathbb{I}(\hat{y}_i \notin V) \quad (11)$$

where $V$ denotes the set of valid semantic IDs.

### A.8 CASE STUDY

In Figure 1b), we further illustrate the reasoning process of our GRPO-enhanced model prior to predicting the next POI. The example shows that the model can effectively associate information from a sequence of SIDs with their corresponding categories, addresses, and related attributes, enabling more accurate next-POI predictions. Compared with previous SID-based approaches, our method allows the model to capture the semantic content embedded in index tokens, thereby improving recommendation accuracy.

### A.9 IMPACT OF HYPERPARAMETERS

Table 3: Impact of validity reward when the predicted semantic ID is valid

| reward value | $Acc@1$ | | $rate_i$ | |
|---|---|---|---|---|
| | NYC | CA | NYC | CA |
| 0.1 | 0.3604 | 0.2267 | 0.58% | 1.26% |
| 0.2 | 0.3618 | 0.2277 | 0.00% | 1.12% |
| 0.3 | 0.3589 | 0.2258 | 0.00% | 1.07% |
| 0.4 | 0.3575 | 0.2247 | 0.00% | 0.98% |
| 0.5 | 0.3502 | 0.2239 | 0.00% | 0.98% |

Inspired by the reward design in DeepSeek-R1(DeepSeek-AI et al., 2025), we assign an accuracy reward of 1.0 when the predicted SID is correct and 0.0 when it is incorrect. Beyond this, we

investigate the effect of introducing a validity reward when the predicted SID is valid. The results, shown in Table 3, indicate that small validity rewards (0.1–0.2) slightly improve Acc@1 in both NYC and CA datasets compared to higher reward values. For instance, setting the reward to 0.2 yields the best performance (0.3618 in NYC and 0.2277 in CA). However, larger rewards (larger than 0.3) lead to a gradual decline in accuracy, suggesting that excessive emphasis on validity may interfere with accuracy optimization. Meanwhile, the invalid prediction rate decreases as the reward increases, with values above 0.3 driving the rate to nearly 0% in NYC and below 1% in CA. This demonstrates a trade-off between improving prediction validity and maintaining overall accuracy.

