# OpenReview forum: "BLCSRec: Bridging Language and Collaborative Semantics for Next POI Recommendation"
_ICLR.cc/2026/Conference — Submitted to ICLR 2026_

### Official Review · Reviewer_uCDa · 2025-10-25

**Soundness:** 2
**Presentation:** 3
**Contribution:** 1
**Rating:** 2
**Confidence:** 5

**Summary:**

The paper proposes a framework that integrates large language models (LLMs) with collaborative filtering for predicting users’ next POI visit. BLCSRec first employs LLMs to generate rich textual profiles for users and POIs that capture spatial, temporal, and behavioral information. It then encodes these profiles into multi-level semantic identifiers using RQ-VAE. To align language and collaborative semantics, the authors introduce explicit (bidirectional mapping between text and SIDs) and implicit (asymmetric prediction) alignment tasks during fine-tuning. Finally, a GRPO reinforcement learning stage with hierarchical rewards penalizes invalid IDs. Experiments demonstrate that the proposed model outperforms several baseline methods.

**Strengths:**

S1. The utilization of LLMs for next POI predictions is a trending topic in research.

S2. The use of RQ-VAE allows the system to translate continuous semantic representations into structured, interpretable identifiers while preserving contextual richness.

S3. Experiments show that BLCSRec achieves the best performance on three datasets.

**Weaknesses:**

W1. The paper’s claimed contributions are incremental and technically weak. The two central challenges mentioned in the paper, namely semantic gap and invalid ID generation, have either been effectively addressed in prior work or can be easily handled through straightforward methods.

- The semantic gap problem has already been explored in earlier studies (e.g., [1]) using multi-objective semantic alignment strategies. The proposed alignment mechanism in this paper follows an almost identical formulation and does not introduce new learning principles or architectural innovations.
- The invalid ID issue can be efficiently handled by simpler deterministic strategies, such as a prefix-constrained decoding or Trie-based masking of invalid token combinations during generation. The adoption of reinforcement learning (GRPO) to penalize invalid IDs appears unnecessary. The framework contains little technical novelty beyond existing methods.

W2. The framework is very similar to an existing study [1], both in methodological design and training objectives. However, the authors fail to clearly acknowledge or differentiate their approach from this closely related study in the Related Work section.

W3. BLCSRec depends heavily on large, high-resource models (e.g., Qwen2.5-7B and GPT-4.1-mini) for both POI profiling and downstream fine-tuning. This design choice makes the framework prohibitively expensive compared to conventional deep learning models, requiring multi-GPU training and long sequence processing. However, the paper provides no discussion of inference efficiency, scalability to real-world settings with massive POIs, or potential deployment feasibility.

\
[1] Enhancing Large Language Models for Mobility Analytics with Semantic Location Tokenization. KDD 2025

**Questions:**

Please refer to comments in Weaknesses.

---

### Official Review · Reviewer_Jui4 · 2025-10-25

**Soundness:** 2
**Presentation:** 3
**Contribution:** 2
**Rating:** 4
**Confidence:** 5

**Summary:**

This paper proposes BLCSRec, a novel framework for next Point-of-Interest (POI) recommendation that addresses two critical challenges in LLM-based approaches using Semantic Identifiers (SIDs): the semantic gap between language and collaborative semantics, and the generation of invalid SIDs. The key contributions include:
(1) POI Profile Generation: Using LLMs to create enriched profiles from user trajectories and POI attributes
(2) Semantic ID Construction: Employing RQ-VAE to encode addresses, categories, and profiles into structured SIDs
(3) Dual Alignment Strategy: Explicit alignment (SID↔text mapping) and implicit alignment (asymmetric prediction tasks) to bridge semantic gaps
(4) GRPO Reinforcement Learning: Hierarchical reward structure to suppress invalid SID generation and improve accuracy

**Strengths:**

1. Integrates profile generation, semantic quantization, dual alignment, and reinforcement learning in a cohesive framework
2. Extensive experiments across multiple datasets with thorough ablation studies and analytical experiments
3. Reduction in invalid SID generation addresses a key deployment concern for real-world systems

**Weaknesses:**

1. The multi-stage training process (SFT + GRPO) with multiple alignment tasks is computationally intensive

2. Limited analysis of semantic understanding

**Questions:**

How sensitive is the performance to the quality of LLM-generated profiles? Have you experimented with different prompting strategies or smaller LLMs for profile generation?

---

### Official Review · Reviewer_ip7a · 2025-10-31

**Soundness:** 2
**Presentation:** 2
**Contribution:** 3
**Rating:** 4
**Confidence:** 4

**Summary:**

This paper introduces BLCSRec, a multi-stage large language model (LLM) fine-tuning framework for next POI (Point of Interest) recommendation. The work pursues two main goals: (1) to enhance the alignment between semantic-token-based and LLM-based next POI models, and (2) to reduce predictions of non-existent POIs.

To address the first goal, the authors propose a series of text-trajectory pretraining-style tasks, comprising explicit tasks (e.g., semantic token-to-POI description mapping) and implicit tasks, where either input or output POI semantic tokens are replaced with their textual descriptions. To address the second goal, they employ GRPO to incentivize the generation of valid and accurate POI semantic tokens.

Experimental results demonstrate that BLCSRec consistently outperforms existing baseline methods, achieving improvements across evaluation metrics.

**Strengths:**

+ The paper introduces a clean and effective approach to improving model alignment through text-trajectory tasks, termed explicit and implicit alignment by the authors. This design helps ensure that semantic tokens acquire more meaningful and robust representations, rather than serving as arbitrary “special token sequences”.

+ The method demonstrates clear performance gains in accuracy compared to the baseline GNPR-SID, which proposed the baseline semantic token + LLM for next POI prediction approach.

+ The ablation study is well-presented, showing consistent improvements from each component of the proposed framework and validating the contribution of each design choice.

**Weaknesses:**

- A major concern lies in the dataset setup, which results in an unfair comparison with baseline methods. In L364-367, the authors stated, “During evaluation, each user’s last visited POI serves as ground truth, while previous visits constitute the input sequence. Notably, during training, users’ historical records are treated as sequential input and concatenated with the test set to a specified length before inference.” However, this setup differs from that of STHGCN (Yan et al., 2023) (L361), where check-ins are grouped into 24-hour trajectories. As such, comparing results obtained from these differing setups is not fair. Ensuring consistent data splitting and experimental setups is essential for a valid comparison.

- The ablation study would be more convincing if results for the TKY dataset were included, particularly since Japanese addresses were translated into English (as mentioned in Appendix A.3).

- Minor presentation issues:
Figure 1 contains a misspelling in its caption.
There is missing whitespace before in-line citations and parentheses.
L75-76 could be better phrased as: “As shown in Figure 1, (a) when .., (b) ..”
In Appendix A.3 (Implementation Details), “GPT-41-mini” appears to be a spelling error, as the said model doesn’t exist.
Paper presentation and clarity can be improved.

**Questions:**

Beyond alignment via text-trajectory pretraining tasks, why were regularization loss terms not considered? For example, semantic token embeddings could potentially be aligned with embeddings in the codebook space instead of relying solely on text-trajectory tasks.

Why was the FSQ-TKY dataset not included in the ablation study? What would the ablation results look like for that dataset?

**Details Of Ethics Concerns:**

No ethical concerns

---

### Official Review · Reviewer_kgEE · 2025-11-01

**Soundness:** 3
**Presentation:** 3
**Contribution:** 2
**Rating:** 4
**Confidence:** 4

**Summary:**

This paper proposes BLCSRec, a framework for next-POI recommendation that bridges the gap between language models and collaborative semantics. The authors design a two-stage learning pipeline consisting of SFT and RFT based on the GRPO algorithm. To improve alignment between semantic IDs and natural language, the paper introduces explicit and implicit alignment objectives during SFT. The method reportedly improves both accuracy and the rate of valid semantic ID generation compared to previous LLM-based baselines.

**Strengths:**

1. The motivation of bridging the semantic gap between textual and collaborative representations is clear and relevant to recent efforts in LLM-based recommendation systems.

2. The use of SFT followed by RFT is a logical design choice, and the paper provides a reasonable explanation of their respective roles. The alignment objectives are well-motivated and seem to address one of the key challenges in applying LLMs to structured ID-based tasks.

3. The paper is generally well written and structured, and the proposed method shows awareness of issues such as invalid ID generation and semantic misalignment.

**Weaknesses:**

1. The most significant issue concerns the reported performance comparison. The GNPR baseline’s results in this submission are much lower than those in its original paper. If the original GNPR results are taken into account, its performance actually exceeds even the best configuration of BLCSRec. This discrepancy raises questions about the fairness and credibility of the experimental results.

2. The RFT stage using GRPO is not entirely convincing. The overall accuracy of around 30 percent indicates that most model outputs during training are likely incorrect or invalid. Since GRPO computes value estimates from grouped responses, a large number of invalid or low-quality generations could lead to sparse and noisy reward signals. This situation makes it difficult to ensure stable and meaningful gradient updates, and the paper does not provide sufficient evidence or analysis to support the effectiveness of the GRPO phase.

3. The paper would benefit from a more detailed ablation or reward analysis to show how GRPO contributes beyond SFT and whether the gains are consistent across multiple runs.

**Questions:**

1. How were the GNPR baseline experiments reproduced? Did you verify your reimplementation against the official checkpoints or scripts?

2. Can the authors provide quantitative analysis or visualization of the GRPO reward distribution during training to confirm that the policy updates are meaningful rather than dominated by noise?

3. Have the authors compared the stability of the RFT process when initialized from different SFT checkpoints?

---

### Meta-Review · Area_Chair_y9oz · 2025-12-22

**Summary:**

This paper proposes BLCSRec, a method for next POI recommendation. Its key motivation is that existing SID-based approaches for next-POI recommendation using LLMs mainly suffer from two issues: (1) the semantic gap between tokenized representations and POI-specific identifiers, and (2) the risk of generating invalid SIDs. The core idea of BLCSRec is to mitigate this semantic gap by linking SIDs with their semantic representations, and it further introduces GRPO to improve accuracy. Overall, the paper’s motivation is clear, the research problem is meaningful, and the presentation is understandable.

However, the following concerns led the paper to receive scores of 2, 4, 4, and 4:

1. There are several concerns regarding the experimental setup and results. For example, the reported performance of the baseline methods differs from that reported in the original papers, the dataset settings may be unfair, and the ablation studies are insufficient. These issues raise concerns about the validity of the experimental evaluation.

2. The proposed method appears to be similar to [1], yet this similarity is not discussed in the related work section.

3. A more detailed comparison and analysis of inference efficiency are required.

All reviewers provided below-acceptance recommendations with high confidence. In addition, the authors did not provide a rebuttal, so these concerns remained unaddressed. After carefully reading all reviewers’ comments and the paper, I believe the current version does not meet the acceptance standard for ICLR.

[1] Enhancing Large Language Models for Mobility Analytics with Semantic Location Tokenization, KDD 2025

**Reviewer Concerns:**

The authors did not provide a rebuttal, so the issues raised by the reviewers remain unresolved. These concerns mainly include:

- potentially unfair experimental settings and results,

- insufficient ablation studies,

- inadequate discussion of inference efficiency,

- limited improvements compared with prior/related work.

**Reviewer Scores:**

I believe it is unlikely that the reviewers will increase their scores, as the paper does not provide a rebuttal and a consensus toward rejection has already been formed.

---

### Decision · Program_Chairs · 2026-01-26

Reject